# Does Cement Augmentation of the Sacroiliac Screw Lead to Superior Biomechanical Results for Fixation of the Posterior Pelvic Ring? A Biomechanical Study

**DOI:** 10.3390/medicina57121368

**Published:** 2021-12-16

**Authors:** Moritz F. Lodde, J. Christoph Katthagen, Clemens O. Schopper, Ivan Zderic, R. Geoff Richards, Boyko Gueorguiev, Michael J. Raschke, René Hartensuer

**Affiliations:** 1AO Research Institute Davos, Clavadelerstrasse 8, 7270 Davos, Switzerland; clemens.schopper@hotmail.com (C.O.S.); ivan.zderic@aofoundation.org (I.Z.); geoff.richards@aofoundation.org (R.G.R.); boyko.gueorguiev@aofoundation.org (B.G.); 2Department for Trauma, Hand and Reconstructive Surgery, University Hospital Münster, Albert-Schweitzer-Campus 1, Building W1, Waldeyerstraße 1, 48149 Münster, Germany; christoph.katthagen@ukmuenster.de (J.C.K.); michael.raschke@ukmuenster.de (M.J.R.); hartensuer@uni-muenster.de (R.H.); 3Department for Orthopaedics and Traumatology, Kepler University Hospital GmbH, Johannes Kepler University Linz, 4040 Linz, Austria

**Keywords:** SI screw, cement augmentation, unstable pelvic ring fracture, biomechanics

## Abstract

*Background and Objectives*: The stability of the pelvic ring mainly depends on the integrity of its posterior part. Percutaneous sacroiliac (SI) screws are widely implanted as standard of care treatment. The main risk factors for their fixation failure are related to vertical shear or transforaminal sacral fractures. The aim of this study was to compare the biomechanical performance of fixations using one (Group 1) or two (Group 2) standard SI screws versus one SI screw with bone cement augmentation (Group 3). *Materials and Methods*: Unstable fractures of the pelvic ring (AO/OTA 61-C1.3, FFP IIc) were simulated in 21 artificial pelvises by means of vertical osteotomies in the ipsilateral anterior and posterior pelvic ring. A supra-acetabular external fixator was applied to address the anterior fracture. All specimens were tested under progressively increasing cyclic loading until failure, with monitoring by means of motion tracking. Fracture site displacement and cycles to failure were evaluated. *Results*: Fracture displacement after 500 cycles was lowest in Group 3 (0.76 cm [0.30] (median [interquartile range, IQR])) followed by Group 1 (1.42 cm, [0.21]) and Group 2 (1.42 cm [1.66]), with significant differences between Groups 1 and 3, *p* = 0.04. Fracture displacement after 1000 cycles was significantly lower in Group 3 (1.15 cm [0.37]) compared to both Group 1 (2.19 cm [2.39]) and Group 2 (2.23 cm [3.65]), *p* ≤ 0.04. Cycles to failure (Group 1: 3930 ± 890 (mean ± standard deviation), Group 2: 3676 ± 348, Group 3: 3764 ± 645) did not differ significantly between the groups, *p* = 0.79. *Conclusions*: In our biomechanical setup cement augmentation of one SI screw resulted in significantly less displacement compared to the use of one or two SI screws. However, the number of cycles to failure was not significantly different between the groups. Cement augmentation of one SI screw seems to be a useful treatment option for posterior pelvic ring fixation, especially in osteoporotic bone.

## 1. Introduction

During the recent two decades the incidence of pelvic ring fractures and the amount of their surgical treatment procedures increased significantly [1]. In unstable pelvic fractures resulting from high-energy trauma, early stable fixation is highly recommended [2]. Fractures resulting from low-energy trauma or physiological stress are classified as fragility fracture of the pelvis (FFP) [3]. Letournel’s golden rule postulates that the reduction and fixation of the weight bearing posterior part is the primary goal in stabilization of the pelvic ring [4,5]. Full weight bearing mobilization and pain reduction are the goals of the treatment of fragility fractures of the pelvis [6,7]. Several treatment options for posterior pelvic ring fractures are described [8,9,10]. Fixation of sacrum fractures with a percutaneous sacroiliac (SI) screw is widely accepted as standard of care treatment of posterior ring fractures [9,11]. However, SI screw loosening occurs in up to 20% of the cases [12,13]. The main risk factors for fixation failure in pelvic ring injuries are related to existence of vertical shear and transforaminal sacral fractures [13]. Consequently, alternative fixation methods for vertical unstable sacral fracture are required [13]. Bone cement augmentation of cannulated SI screws characterized by additional perforations at the screw tip is described clinically as a successful adjunct method [9,14] (Figure 1). 

A previous biomechanical study reported enhanced anchorage at the distal screw end by cement augmentation [15]. The biomechanical results from another study confirmed that cement augmentation of the SI screw in the S1 and S2 sacral bodies may have a positive effect [16]. The use of two SI screws placed in the S1 and S2 bodies represent a further alternative fixation method [12,13,16,17,18]. However, SI screws placed in the S1 and S2 bodies require an appropriate transsacral corridor [16]. In approximately 30% of the cases this corridor is not present [19]. Biomechanical studies exploring the effect of different SI screw techniques are still required [16]. 

Therefore, the aim of the present biomechanical study was to compare the biomechanical performance of fixations of vertical unstable sacral fractures using one or two standard SI screws versus one SI screw with cement augmentation. It was hypothesized that augmentation of the SI screw would result in its biomechanical superiority compared to the other fixation methods.

## 2. Materials and Methods

Unstable fractures on the right side of the pelvic ring (AO/OTA 61-C1.3, FFP IIc) were simulated in 21 artificial pelvises (Model #LS4060, Synbone, Zizers, Switzerland) (Figure 2).

An anterior pelvic ring fracture was created via transverse cuts of the superior and inferior ramus of the pubis, set 2 cm laterally to the pubic tubercle. A posterior pelvic ring fracture was simulated by a paraforaminal transverse osteotomy of the sacrum through the midline between the medial osseous frontier of the SI joint and the lateral osseous frontier of the first anterior sacral foramen in zone 1, according to Denis et al. [20]. All osteotomies were set using a customised cutting template ensuring consistent course of fracture lines [21]. The pelvises were assigned to three groups of 7 specimens each for application of the following fixation methods: one (Group 1) or two (Group 2) standard SI screws and one SI screw with additional cement augmentation (Group 3) (Figure 3). 

A supra-acetabular 5.0 mm external fixator (DePuy Synthes, Zuchwil, Switzerland) was applied to address the anterior fracture in all specimens. A polymethylmethacrylate- (PMMA, SCS-Beracryl, Suter-Kunststoffe AG, Fraubrunnen, Switzerland) aiming guide was used to achieve best-possible instrumentation reproducibility. The standardised customised PMMA guide was used to predrill the entry points of the 5.0 mm Schanz screws with a 3.5 mm drill bit. The Schanz screws were inserted over the entire length of their thread. Insertion of the SI screws was performed according to the standard percutaneous operation technique [22,23]. Predrilling for the SI screw insertion was performed with a 5.0 mm drill bit using customised PMMA guides for the S1 and S2 corridors. Cannulated fully threaded 7.5 × 90 mm titanium SI screws (Axomed GmbH, Freiburg, Germany) were inserted for fixation of the posterior ring of the specimens. All screws were tightened at 1.5 Nm using an electronic torque screwdriver (PB 8320 A 0.4–2.5, PB Swiss Tools, Wasen/Bern, Switzerland). Augmentation of the cannulated fully threaded 7.5 × 90 mm screws was performed using 3 ml Traumacem V+ bone cement (DePuy Synthes, Zuchwil, Switzerland) prepared according to the manufacturer’s guidelines [14]. The injected cement spread around the perforating holes at the distal end of the SI screws (Figure 3C).

Optical markers were glued on the medial and lateral aspects of the fracture site of each sacrum and at the right SI joint for motion tracking (Figure 2 and Figure 3).

### 2.1. Biomechanical Testing

Biomechanical testing was performed on an electrodynamic material testing machine (Acumen, MTS Systems Corp., Eden Prairie, MN, USA) equipped with a 3.0 kN load cell in a setup simulating one-legged stance with applied load at the right hemipelvis (Figure 4) [15,16,24,25].

By using a unipolar hemiarthroplasty attached to a PMMA-potted acetabular cup—the latter being press—fit fixed to the bone model-standardisation of the hip joint loading mechanics was achieved. Cranially, each central body of the sacrum was fixed with two screws plus washers, applied through the first row of neuroforamina within the sacral body, and a PMMA cast, to an L-shaped profile featuring a posterior section made of cotton laminates (Canevasite, HBW 2088, Amsler and Frey AG, Schinznach-Dorf, Switzerland) [21]. The L-shaped frame was fixed to the load cell and the machine transducer. Due to the alignment of the specimens to the machine axis, an axial compression force was applied through the centre of the S1 vertebral body [15]. Caudally, the hemiarthroplasty was rigidly constrained to the machine base via a socket brace.

Subsequently, applying a sinusoidal loading profile of each cycle, the specimens were tested under progressively increasing cyclic loading at 2 Hz. Starting from 50 N, the peak load of each cycle was increased at a rate of 0.05 N/cycle, whereas its valley load was kept at a constant level of 20 N. The test was stopped when a distinct failure of the bone-implant construct was observed, or the machine actuator reached 30 mm displacement.

### 2.2. Data Acquisition and Analysis

Cycles to failure were evaluated retrospectively analysing machine data with regard to the test stop criteria. Interfragmentary movements were continuously measured in all six degrees of freedom using two optical cameras (ARAMIS SRX, GOM GmbH, Braunschweig, Germany) at a rate of 50 Hz. The measurement sensitivity of the marker locations was ±0.004 mm in the XY plane (frontal to the cameras) and along the *z*-axis (depth) [24]. A local coordinate system of the posterior osteotomy was defined by its *x*-axis oriented normally to the osteotomy plane, with *y*-axis and *z*-axis lying vertically and horizontally in the osteotomy plane, respectively. Fracture displacement was measured on both fracture sites as the magnitude of their corresponding three-dimensional translation with respect to each other. It was evaluated after 500 and 1000 cycles relative to the initial specimen’s state at the test start under corresponding peak loading conditions. Statistical analysis was performed using IBM SPSS Statistics (v.23, IBM, Armonk, NY, USA). Normality of data distribution was checked with the Shapiro–Wilk test. Based on the data distribution, either parametric or non-parametric tests were conducted to screen for significant differences among the groups using either One-Way Analysis of Variance (ANOVA) or Kruskal–Wallis tests with Bonferroni correction for multiple comparisons. Level of significance was set at 0.05 for all statistical tests.

## 3. Results

*p*-Values from pairwise comparisons between the groups for fracture displacement after 500 and 1000 cycles are summarized in Table 1. At both time points fracture displacement was significantly lower for specimens treated with augmented SI screw compared to one standard SI screw, *p* = 0.04 (Figure 5 and Figure 6). In addition, whereas fracture displacement after 500 cycles was only trend wise lower for fixation with augmented SI screw versus two standard SI screws (*p* = 0.08), this difference became significantly different after 1000 cycles (*p* = 0.02). No significant differences were detected between fixations using one standard versus two standard SI screws for both time points (*p* = 0.99).

Cycles to failure for the fixations using one standard SI screw (3930 ± 890 (mean ± standard deviation)), two standard SI screws (3676 ± 348) and one cement augmented SI screw (3764 ± 645) demonstrated no significant difference among the groups (*p* = 0.79). The failure mode for all specimens was similar and featured failure of the anterior and posterior pelvic ring when the machine transducer reached 30 mm displacement.

## 4. Discussion

The present study explored the biomechanical competence of one SI screw, two SI screws and of one cement augmented SI screw for fixation of an unstable pelvic ring fracture. Cement augmentation of the SI screw led to significantly less fracture displacement and micromovements at the posterior pelvic ring. However, the number of cycles to failure was not significantly different among the groups. The anchorage of the SI screw at the lateral aspect of the Os ilium and SI joint, and the additional firm anchorage of the cement cloud around the SI screw tip might lead to a more stable fixation and thus to less fracture displacement and micromovements. 

Previous biomechanical studies evaluating posterior pelvic ring fixation techniques reported biomechanical advantages for cement augmentation compared to single SI screw fixation only [15,16,26] in terms of significantly higher stiffness and pull-out force [27]. Augmentation of SI screws increases their pull-out strength [15,27]. However, regarding overall construct stability, no significant difference between augmented and non-augmented screws was reported due to washer penetration in the iliac bone [15]. 

This could have been the reason indicating the comparable biomechanical competence of the three tested techniques in the current study regarding their number of cycles to failure—a result which is in accordance with previous biomechanical data [16,24]. Significant differences in fracture displacement were measured over the time points as previously described [24]. In the current study, fracture displacement after 500 and 1000 cycles was significantly less in the augmented SI screw group compared to both groups using one or two standard SI screws. This reduced movement may clinically result in less implant loosening.

Interestingly, previous biomechanical data demonstrated improved stability using two lateral fixation points for unstable pelvic ring fractures [28,29]. In the present study, no biomechanical advantage could be observed when inserting two SI screws. Possible reasons for these different reported results might be the substantially different test setups used in the previous studies [28,29], their loading arrangements [28] and the fact that the pubic symphysis was not fixed as it might be in a clinical setting [28]. A further possible reason is that the SI screw used in the previous studies had no washer [29] and hence was anchored less firmly in the ilium. Furthermore, it might be indicated that a bilateral implant bone anchorage results in higher stability than two SI screws with only unilateral firm bone implant anchorage. In the present study the presence of bilateral bone implant anchorage was significantly more important regarding fracture displacement than the number of inserted SI screws.

Clinically, the SI screw showed higher biomechanical stability compared to other fixation methods [30,31]. Both SI screw and augmented SI screw are well-accepted treatment options with a low complication rate [9] for treatment of posterior pelvic ring fractures. The minimally invasive approach and the fact that the insertion of a SI screw is almost always feasible are further benefits [32]. However, Kim et al. observed a SI screw loosening in 17.3% of the cases [13] and further differentiated between simple back-out and failure of the SI screw. In 11.8% of the cases failure of the SI screw occurred, which led to revision surgery. In contrast, the simple back-out mechanism resulted in bone healing without any further surgical procedure in the remaining cases [13]. Griffin et al. [33] analysed retrospectively 62 patients with vertical unstable pelvic ring fractures treated with SI screw. A reason for the observed SI screw loosening was the lever-arm mechanism with simultaneous existing poor bone quality [24,33]. Augmentation of the SI screw inhibits the action of this lever-arm mechanism and leads to higher stiffness and pull-out forces [27]. A further clinical study analysing the possible benefit of the cement augmented SI screw, related to the bilateral screw anchorage in the bone inhibiting the action of the lever-arm mechanism, would be of interest. A recent clinical study could observe such advantages as a shorter hospital stay of patients treated with cement augmented versus non-augmented SI screws [9]. Cement-associated complications (mainly radiological) were seen in 22% of the cases [9], but cement augmentation was not associated with increased specific or neurological clinical complications. More clinical studies are necessary to further evaluate possible advantages of SI screw augmentation compared to the conventional SI screw technique. 

In addition, the successful use of an additional SI screw inserted in the S2 sacral body to strengthen the fixation of posterior pelvic ring fractures is described in literature [12,17,18]. However, from a clinical point of view, insertion of only one SI screw may be favourable over insertion of two screws due to a smaller approach and corridor [19], shorter surgery and radiation times, and less number of used implants. 

Due to the large displacements in the model used in the current study, optical motion tracking was efficient only up to 1000 test cycles, because the fracture displacement was too big for capturing by the system cameras afterwards. Cycles to failure were analysed up to 4000 cycles, which is comparable to previous biomechanical studies [15,16,24,30]. Further limitations of this study are similar to those inherent to all biomechanical studies using synthetic bones. Outcomes from biomechanical testing using synthetic bones without soft tissue, ligaments and muscles may differ from those when using human cadaveric specimens [34,35,36]. However, previous reports conclude that synthetic bones represent an appropriate replacement for cadaver specimens [35,36,37]. Using artificial bone models has the advantage of similarity in material characteristics between samples [16]. Additionally, reliability of the conducted procedures was achieved using standardized methods such as individually customized PMMA guides for osteotomy setting and implantation (Figure A1, Figure A2 and Figure A3) [21]. Single-legged stance was chosen for testing as reported in previous studies [16,38] to simulate appropriately the clinical situation [25,39]. The continuous measurement of the fracture displacement in all six degrees of freedom using two optical cameras of a very precise motion tracking system is a further strength of the present study (Figure A4). 

## 5. Conclusions

In our biomechanical setup cement augmentation of one SI screw resulted in significantly less displacement compared to the use of one or two SI screws. However, the number of cycles to failure was not significantly different between the groups. Cement augmentation of one SI screw seems to be a useful treatment option for posterior pelvic ring fixation, especially in osteoporotic bone.

## Figures and Tables

**Figure 1 medicina-57-01368-f001:**
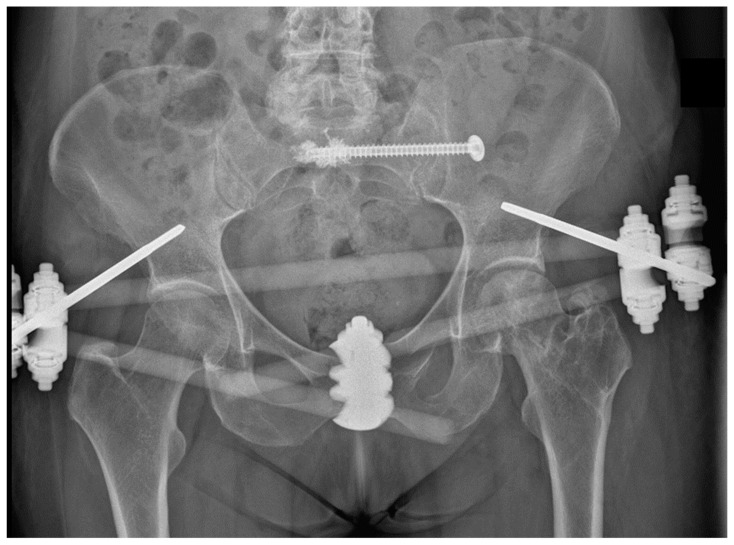
Antero-posterior radiograph demonstrating SI screw fixation of the posterior pelvic ring in a 67-year-old patient who sustained a AO/OTA 61-C1.3, FFP IIc fracture. A bilateral implant bone anchorage was achieved by cement augmentation around the tip of the SI screw and by a screw washer at the ilium. The anterior pelvic ring was addressed with a supra-acetabular external fixator.

**Figure 2 medicina-57-01368-f002:**
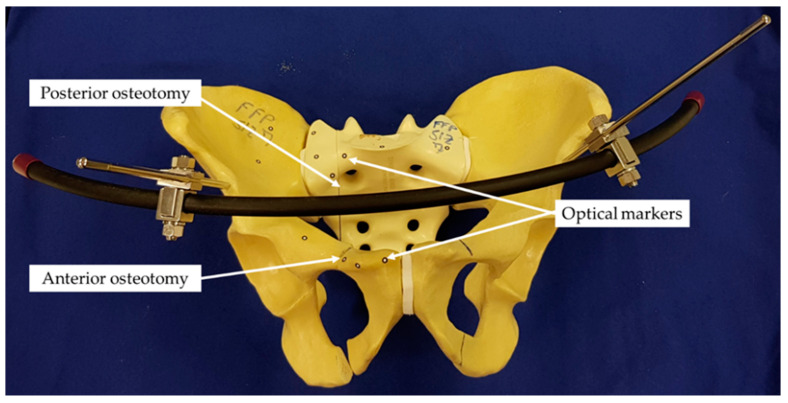
Anterior view of a specimen prepared for biomechanical testing with anterior and posterior osteotomies simulating an unstable pelvic ring fracture and equipped with optical markers for motion tracking. The anterior pelvic ring was addressed with a supra-acetabular external fixator.

**Figure 3 medicina-57-01368-f003:**
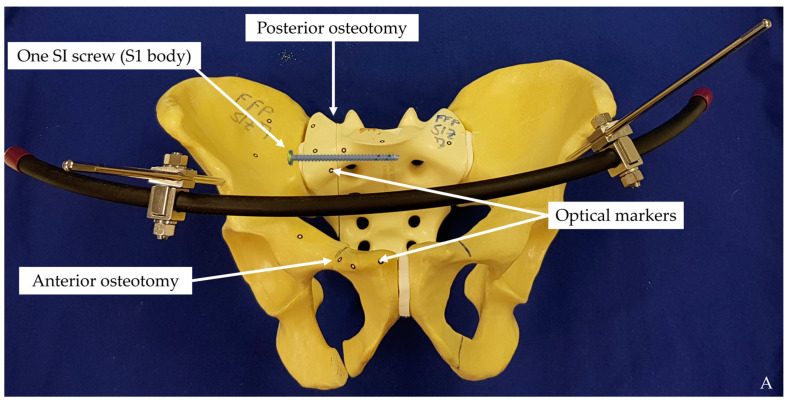
Exemplified specimens from each study group instrumented with either one SI screw (S1 body) ((**A**), Group 1), two SI screws (S1 body and S2 body) ((**B**), Group 2) and one augmented SI screw (S1 body) ((**C**), Group 3).

**Figure 4 medicina-57-01368-f004:**
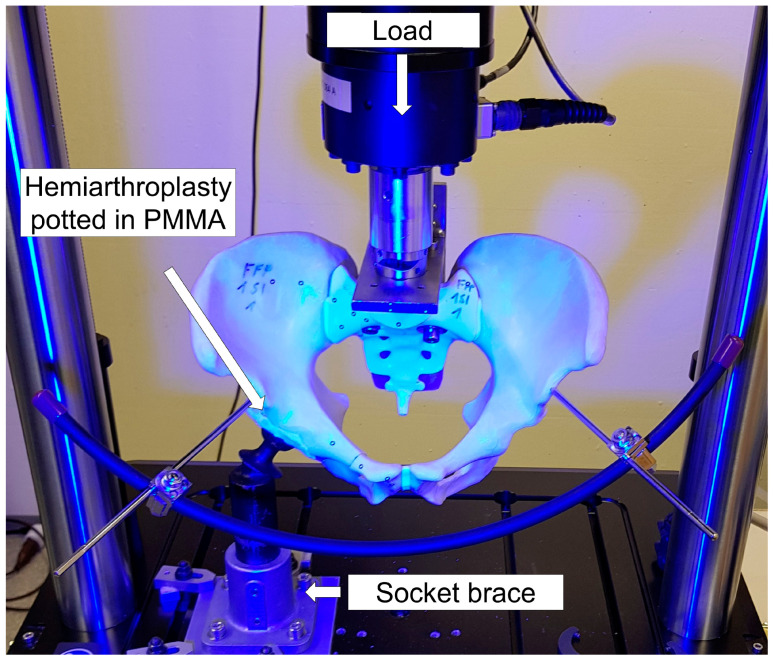
Setup with a specimen mounted for biomechanical testing in one-legged stance position.

**Figure 5 medicina-57-01368-f005:**
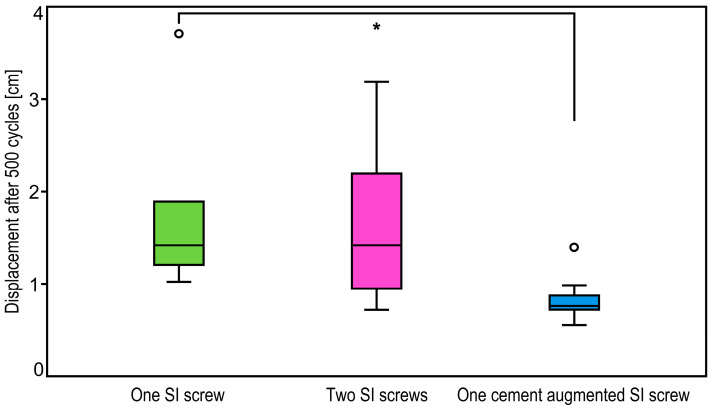
Boxplots showing fracture displacement in the groups after 500 cycles with a star indicating significant difference between the treatments using one cement augmented SI screw or one SI screw.

**Figure 6 medicina-57-01368-f006:**
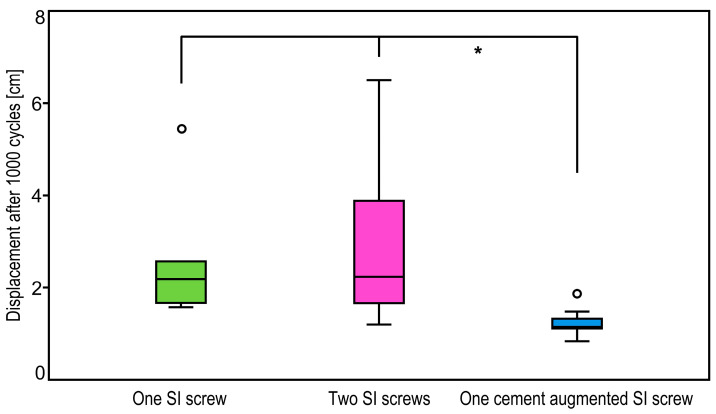
Boxplots showing fracture displacement in the groups after 1000 cycles with a star indicating significant differences between the treatment with one cement augmented SI screw and both treatments using one or two SI screws.

**Table 1 medicina-57-01368-t001:** *p*-Values from pairwise comparisons of fracture displacement after 500 and 1000 cycles.

Cycles	Pairwise Group Comparisons	*p*-Value
500	One SI screw vs. two SI screws	0.99
	One SI screw vs. one cement augmented SI screw	0.04
	Two SI screws vs. one cement augmented SI screw	0.08
1000	One SI screw vs. two SI screws	0.99
	One SI screw vs. one cement augmented SI screw	0.04
	Two SI screws vs. one cement augmented SI screw	0.02

## Data Availability

Data are contained within the article.

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
