# Peer review of "Does Cement Augmentation of the Sacroiliac Screw Lead to Superior Biomechanical Results for Fixation of the Posterior Pelvic Ring? A Biomechanical Study"

_medicina, 2021, doi:10.3390/medicina57121368_

Round 1

Reviewer 1 Report

This is a well-conducted and well-written study approaching once again the topic of posterior pelvic ring fixation. As a pelvic surgeon, I appreciate reading the manuscript even if biomechanical tests on specimens have previously shown limitations in their clinical application, especially in the field of pelvic fracture. However, the authors have been clear about that describing the limitations of the study.

I may report a few observations.

Many forces in different directions influence the screw strength (Toggling, pull-out, rotation, and vertical share for example). Among those, it is my idea that the pull-out strength of the screws should be approached because proportional to the compression in the site of fracture.  Several authors reported that it is fundamental to the healing process and limits the screw failing under vertical shearing forces due to the increase in friction in the site of fracture. So, does the augmentation improve the pull-out strength?

Concluding, maybe the Authors could approach better the results in the discussion suggesting the reason why the augmentation showed better results? is it possible that screw head implanted in harder tissue (cement VS osteoporotic cancellous bone) had less toggling? could the pull-out strength play a role?

thank you.

Author Response

Response to Reviewer 1 Comments

Dear Editor,

thank you for your kind letter concerning our manuscript. We were happy to see the encouraging comments and wise instructions by the reviewers. Below, we comment every question and suggestion and show the changes we have made in the revised manuscript that is enclosed.

This is a well-conducted and well-written study approaching once again the topic of posterior pelvic ring fixation. As a pelvic surgeon, I appreciate reading the manuscript even if biomechanical tests on specimens have previously shown limitations in their clinical application, especially in the field of pelvic fracture. However, the authors have been clear about that describing the limitations of the study.

  1. Many forces in different directions influence the screw strength (Toggling, pull-out, rotation, and vertical share for example). Among those, it is my idea that the pull-out strength of the screws should be approached because proportional to the compression in the site of fracture. Several authors reported that it is fundamental to the healing process and limits the screw failing under vertical shearing forces due to the increase in friction in the site of fracture. So, does the augmentation improve the pull-out strength?
  2. Thank you for raising this question. From our clinical experience augmentation of the sacroiliac screw is effective and especially in osteoporotic bone advantageous. Augmentation of the screw increases the pull-out strength and leads to more load cycles until loss of fixation. Furthermore, in a biomechanical study augmentation was associated with less pull-out failures.

Please see lines: 231-237, 270-271

Safety, Effect and Feasibility of Percutaneous SI-Screw with and without Augmentation-A 15-Year Retrospective Analysis on over 640 Screws. Hartensuer R, Lodde MF, Keller J, Eveslage M, Stolberg-Stolberg J, Riesenbeck O, Raschke MJ. J Clin Med. 2020 Aug 17;9(8):2660. doi: 10.3390/jcm9082660

Biomechanical comparison of augmented versus non-augmented sacroiliac screws in a novel hemi-pelvis test model. Grüneweller N, Raschke MJ, Zderic I, Widmer D, Wähnert D, Gueorguiev B, Richards RG, Fuchs T, Windolf M. J Orthop Res. 2017 Jul;35(7):1485-1493. doi: 10.1002/jor.23401

Grechenig, S.; Gänsslen, A.; Gueorguiev, B.; Berner, A.; Müller, M.; Nerlich, M.; Schmitz, P. PMMA-augmented SI screw: a biomechanical analysis of stiffness and pull-out force in a matched paired human cadaveric model. Injury 2015, 46, S125-S128, doi:10.1016/S0020-1383(15)30031-0

  1. Concluding, maybe the Authors could approach better the results in the discussion suggesting the reason why the augmentation showed better results? is it possible that screw head implanted in harder tissue (cement VS osteoporotic cancellous bone) had less toggling? could the pull-out strength play a role?
  2. Thank you again for highlighting this important topic. From our point of view augmentation showed better results due to the bilateral screw anchorage in the bone reducing the lever-arm mechanism. As a consequence, the better anchorage lead to less toggling and increases the pull-out strength.

Safety, Effect and Feasibility of Percutaneous SI-Screw with and without Augmentation-A 15-Year Retrospective Analysis on over 640 Screws. Hartensuer R, Lodde MF, Keller J, Eveslage M, Stolberg-Stolberg J, Riesenbeck O, Raschke MJ. J Clin Med. 2020 Aug 17;9(8):2660. doi: 10.3390/jcm9082660

Biomechanical comparison of augmented versus non-augmented sacroiliac screws in a novel hemi-pelvis test model. Grüneweller N, Raschke MJ, Zderic I, Widmer D, Wähnert D, Gueorguiev B, Richards RG, Fuchs T, Windolf M. J Orthop Res. 2017 Jul;35(7):1485-1493. doi: 10.1002/jor.23401

Grechenig, S.; Gänsslen, A.; Gueorguiev, B.; Berner, A.; Müller, M.; Nerlich, M.; Schmitz, P. PMMA-augmented SI screw: a biomechanical analysis of stiffness and pull-out force in a matched paired human cadaveric model. Injury 2015, 46, S125-S128, doi:10.1016/S0020-1383(15)30031-0

Please see lines: 254-256

Reviewer 2 Report

It is an interesting study but the results at 500 and 1000 cycles are clinically not significant. Also the 500 and 1000 cycles are clinically not significant. There is no difference among implants at cycles to failure. All groups failed between 3000-4000 cycles without a difference.

Basically the authors have to conclude that there is no clinically relevant difference among these experimental set-ups.

Author Response

Response to Reviewer 2 Comments

Dear Editor,

thank you for your kind letter concerning our manuscript. We were happy to see the encouraging comments and wise instructions by the reviewers. Below, we comment every question and suggestion and show the changes we have made in the revised manuscript that is enclosed.

  1. It is an interesting study but the results at 500 and 1000 cycles are clinically not significant. Also the 500 and 1000 cycles are clinically not significant. There is no difference among implants at cycles to failure. All groups failed between 3000-4000 cycles without a difference. Basically the authors have to conclude that there is no clinically relevant difference among these experimental set-ups.
  2. Thank you for raising this important topic regarding biomechanical studies. In the present biomechanical set-up significant differences were found at 1000 cycles. The aim of the present biomechanical set-up is to compare different fixation techniques for posterior pelvic ring fractures. Our results show that in the present biomechanical study augmentation leads to significantly less displacement. We have changed the limitations and discuss the general problem of biomechanical studies more specific. please see lines 285-297: “

Furthermore, we discuss it in our discussion referring to the results of previous biomechanical studies, please see lines 231-237: “Previous biomechanical studies evaluating posterior pelvic ring fixation techniques also showed biomechanical advantages for cement augmentation compared to single SI screw fixation only [15,16,26] and a significantly higher stiffness and pull-out force [27]. However, regarding overall construct stability no significant difference between aug-mented and non-augmented screws was described because of the penetration of the washer at the iliac bone [15].”

We have changed our conclusions, please see lines 299-303: “From a biomechanical perspective, in the present model cement augmentation of the SI screw was biomechanically superior compared to the standard SI screw and the S1-S2 screw fixations. The augmentation of the SI screw seems to increase bone-implant anchorage and therefore presents a successful alternative for posterior pelvic ring fixation especially in osteoporotic bone.”

We now are underlining that the results are applicable mainly to the present biomechanical set-up and that significant differences were found regarding the displacement. We conclude that augmentation of the SI-screw seems to increase bone implant anchorage.

Elfar, J.; Menorca, R.M.G.; Reed, J.D.; Stanbury, S. Composite bone models in orthopaedic surgery research and education. J. Am. Acad. Orthop. Surg. 2014, 22, 111–120, doi:10.5435/JAAOS-22-02-111.

O'Neill, F.; Condon, F.; McGloughlin, T.; Lenehan, B.; Coffey, C.; Walsh, M. Validity of synthetic bone as a substitute for osteoporotic cadaveric femoral heads in mechanical testing: A biomechanical study. Bone Joint Res. 2012, 1, 50–55, doi:10.1302/2046-3758.14.2000044.

Wähnert, D.; Hoffmeier, K.L.; Klos, K.; Stolarczyk, Y.; Fröber, R.; Hofmann, G.O.; Mückley, T. Biomechanical characterization of an osteoporotic artificial bone model for the distal femur. J. Biomater. Appl. 2012, 26, 565–579, doi:10.1177/0885328210378057.

Reed, J.D.; Stanbury, S.J.; Menorca, R.M.; Elfar, J.C. The emerging utility of composite bone models in biomechanical studies of the hand and upper extremity. J. Hand Surg. Am. 2013, 38, 583–587, doi:10.1016/j.jhsa.2012.12.005

Round 2

Reviewer 2 Report

The authors have not addressed my concerns. 

The number of cycles are not clinically significant. it is less then an hour of walking. with crutches. 

There is no significant difference to failure. The statistical significance at 1000 cycles is clinically insignificant.

The conclusions are not reflections of the findings. This is a negative study and the authors have to report it accordingly. Otherwise this will lead disasters and false security in clinical setting and irresponsibly driving industry. 

Author Response

Reviewer 2:

The authors have not addressed my concerns. The number of cycles are not clinically significant. it is less then an hour of walking with crutches. There is no significant difference to failure. The statistical significance at 1000 cycles is clinically insignificant. The conclusions are not reflections of the findings. This is a negative study and the authors have to report it accordingly. Otherwise this will lead disasters and false security in clinical setting and irresponsibly driving industry.

Dear reviewer,

Thank you for pointing out that previous editing did not result in the intended improvement of quality. Thank you for the opportunity for further revisions.

We did not intend to indicate any fact not based on the results of our study and we have therefore altered the text to specify that there was no difference regarding cycles to failure. Please see ll. 31-36, 222-223, 237-238, 304-308.

We also found that fracture movement after 500 (compared to single non augmented screw) and after 1000 (compared to both other groups) was significantly less in the augmented group. We agree that differences after 1000 cycles may not reflect a clinical important period. Please see l. 288-289.

However, the numbers of cycles are comparable to the lower limits of published protocols:

  • Biomechanical comparison of augmented versus non-augmented sacroiliac screws in a novel hemi-pelvis test model Grüneweller et al. https://doi.org/10.1002/jor.23401
  • Cement Augmentation in Sacroiliac Screw Fixation Offers Modest Biomechanical Advantages in a Cadaver Model Osterhoff et al. 10.1007/s11999-016-4934-9
  • Screw-in-screw fixation of fragility sacrum fractures provides high stability without loosening-biomechanical evaluation of a new concept. Zderic et al. doi:10.1002/jor.24895
  • Biomechanical investigation of four different fixation techniques in sacrum Denis type II fracture with low bone mineral density Acklin et al. https://doi.org/10.1002/jor.23798

Nevertheless, biomechanical superiority is speculative at this time, and we have edited the text to state that our conclusion is only suggested by our results. 

Several statements that we made were more ambiguous than intended, and we have adjusted the text to be clearer. We hope that our findings of reduced early micromovement will be one additional part of the puzzle in understanding the potential effect of cement augmentation.

We also hope that our changes will now address the reviewer’s concerns. We again thank the reviewer for the valuable advice. This has definitively improved the quality of our work and will hopefully result in consecration for publication.

Yours sincerely,

The authors